# Oxidation of difluorocarbene and subsequent trifluoromethoxylation

Jiao Yu[1], Jin-Hong Lin[1], Donghai Yu[1], Ruobing Du[1] & Ji-Chang Xiao[1]*

As a versatile intermediate, difluorocarbene is an electron-deficient transient species, meaning that its oxidation would be challenging. Herein we show that the oxidation of difluorocarbene could occur smoothly to generate carbonyl fluoride. The oxidation process is confirmed by successful trifluoromethoxylation, $^{18}$O-trifluoromethoxylation, the observation of AgOCF$_3$ species, and DFT calculations.

[1] Key Laboratory of Organofluorine Chemistry, Shanghai Institute of Organic Chemistry, University of Chinese Academy of Sciences, Chinese Academy of Sciences, 345 Lingling Road, 200032 Shanghai, China. *email: jchxiao@sioc.ac.cn

D ue to the unique properties of fluorine element such as strong electronegativity and small atomic radius, the incorporation of fluorine atom(s) into organic molecules could usually lead to profound changes of the latter's physical, chemical, and biological properties[1]. Therefore, significant efforts have been directed towards the development of efficient methods for introducing fluorine or fluorinated moieties into organic compounds[2,3]. Difluorocarbene (:CF$_2$) has served as a versatile intermediate and the transformations of difluorocarbene has proved to be quite efficient for fluorine incorporation[4,5]. Typical difluorocarbene conversions, including insertions into X-H bonds (X=O, N, S, etc.)[4,6,7], [2 + 1] cycloadditions with multi-bonds[8,9], and coupling with other carbenes[10–12], can conveniently construct various fluorinated functionalities, such as difluoromethyl, *gem*-difluorocyclopropyl and *gem*-difluoroalkenyl groups. However, these typical reactions are limited to the incorporation of a -CF$_2$- moiety. We have previously found that difluorocarbene is so reactive that it can be readily trapped by a suitable sulfur[13–15], selenium[16], or nitrogen source[17] to generate thiocarbonyl fluoride (CF$_2$=S), selenocarbonyl fluoride (CF$_2$=Se), and cyanide anion (CN$^-$), respectively (Fig. 1a–c). On the basis of these findings, which offers more possibilities for difluorocarbene chemistry, it is reasonable to conceive that the oxidation of difluorocarbene with a suitable oxygen source may proceed to afford carbonyl fluoride (CF$_2$=O) (Fig. 1d). Usually, oxidation reactions could proceed smoothly to oxidize electron-rich substrates, but not to electron-deficient substrates[18,19]. Since difluorocarbene is an electron-deficient transient intermediate[20], its oxidation would be a challenging task. Furthermore, because CF$_2$=O is a highly reactive gas and thus hard to detect, it cannot be determined simply by spectroscopic monitoring of the reaction whether the oxidation process occurs or not.

Herein we describe the oxidation of difluorocarbene by using diphenyl sulfoxide (Ph$_2$S=O) as the oxidant to provide carbonyl fluoride, a process which is confirmed by successful trifluoromethoxylation and $^{18}$O-trifluoromethoxylation reactions, the observation of AgOCF$_3$ species, and DFT calculations. A late-stage trifluoromethoxylation for the synthesis of a Trioxsalen derivative is shown to further demonstrate the synthetic utility of this trifluoromethoxylation protocol.

## Results

**Optimization of the trifluoromethoxylation conditions.** Ph$_3$P$^+$CF$_2$CO$_2^-$, developed by us recently[21], and AgF were used as a difluorocarbene reagent and the fluoride source, respectively, in our efforts to ascertain the oxidation process via the trifluoromethoxylation of benzyl bromide **1-1** (Table 1). AgF was used to convert CF$_2$=O into AgOCF$_3$, which may be experimentally observed[22] to support the oxidation process. The oxidants were initially screened, but no desired trifluoromethoxylation product was detected in most cases (Table 1, entries 1–5). To our delight, the use of DMSO (dimethyl sulfoxide) as the oxidant afforded the expected product in 9% yield (Table 1, entry 6), suggesting that sulfoxides may be a suitable class of oxidants. We then examined other sulfoxides (Table 1, entries 7–8) and diphenyl sulfoxide was found to be a superior choice (Table 1, entry 8). Other fluoride sources, including inorganic (Table 1, entries 9–11) and organic (Table 1, entry 12, TBAF=tetra-*n*-butylammonium fluoride) fluoride salts, were examined, but they were all ineffective. This indicates that the Ag ion may play an important role in the reaction. A brief survey of reaction solvents (Table 1, entries 13–17) showed that THF (tetrahydrofuran) or DCM (dichloromethane) was the suitable solvent for this conversion (Table 1, entries 15 and 16). The use of 2,2′-bipyridine or a crown ether as a ligand (Table 1, entries 18

and 19) significantly increased the product yield. A 67% yield was obtained if both bipyridine and the crown ether were present (Table 1, entry 20). The concentration affected the reaction slightly, and the yield increased with increasing concentration (Table 1, entry 21 vs entry 20). At this concentration, the yield decreased if either the crown ether or 2,2′-bipyridine was not used (Table 1, entries 22−23).

**Fig. 1** The transformations of difluorocarbene. **a** The transformation of difluorocarbene into thiocarbonyl fluoride. **b** The transformation of difluorocarbene into selenocarbonyl fluoride. **c** The transformation of difluorocarbene into cyanide anion. **d** The transformation of difluorocarbene into carbonyl fluoride.

**Table 1 Optimization of trifluoromethoxylation conditions.**

| Entry | [O] | [F$^-$] | 1-1:2:3:4$^a$ | Solvent | Yield (%)$^b$ |
|---|---|---|---|---|---|
| 1 | **3a** | AgF | 1:2:2:2 | CH$_3$CN | ND |
| 2 | **3b** | AgF | 1:2:2:2 | CH$_3$CN | ND |
| 3 | **3c** | AgF | 1:2:2:2 | CH$_3$CN | ND |
| 4 | **3d** | AgF | 1:2:2:2 | CH$_3$CN | ND |
| 5 | **3e** | AgF | 1:2:2:2 | CH$_3$CN | ND |
| 6 | **3f** | AgF | 1:2:2:2 | CH$_3$CN | 9 |
| 7 | **3g** | AgF | 1:2:2:2 | CH$_3$CN | 9 |
| 8 | **3h** | AgF | 1:2:2:2 | CH$_3$CN | 24 |
| 9 | **3h** | NaF | 1:2:2:2 | CH$_3$CN | ND |
| 10 | **3h** | KF | 1:2:2:2 | CH$_3$CN | ND |
| 11 | **3h** | CsF | 1:2:2:2 | CH$_3$CN | ND |
| 12 | **3h** | TBAF | 1:2:2:2 | CH$_3$CN | ND |
| 13 | **3h** | AgF | 1:2:2:2 | DMF | 15 |
| 14 | **3h** | AgF | 1:2:2:2 | DMSO | ND |
| 15 | **3h** | AgF | 1:2:2:2 | THF | 33 |
| 16 | **3h** | AgF | 1:2:2:2 | DCM | 32 |
| 17 | **3h** | AgF | 1:2:2:2 | NMP | 14 |
| 18$^c$ | **3h** | AgF | 1:2.5:2:2 | THF | 55 |
| 19$^d$ | **3h** | AgF | 1:2.5:2:2 | THF | 52 |
| 20$^e$ | **3h** | AgF | 1:2.5:2.5:2 | THF | 67 |
| 21$^{ef}$ | **3h** | AgF | 1:2.5:2.5:2 | THF | 74 |
| 22$^{fg}$ | **3h** | AgF | 1:2.5:2.5:2 | THF | 66 |
| 23$^{fh}$ | **3h** | AgF | 1:2.5:2.5:2 | THF | 51 |

Reaction conditions: **1-1** (0.2 mmol), Ph$_3$P$^+$CF$_2$CO$_2^-$, [O], [F$^-$] in solvent (2 mL) at 60 °C for 0.5 h
ND, not detected
$^a$Molar ratio
$^b$Yields were determined by $^{19}$F NMR spectroscopy
$^c$2,2′-Bipyridine (1 equiv) was used as a ligand
$^d$2,3,11,12-Dibenzo-18-crown-6 (1 equiv) was used as a ligand
$^e$2,2′-Bipyridine (1.5 equiv) and 2,3,11,12-dibenzo-18-crown-6 (0.5 equiv) were used
$^f$THF (1.5 mL) was used
$^g$2,2′-Bipyridine (1.5 equiv) was used without the crown ether.
$^h$2,3,11,12-Dibenzo-18-crown-6 (0.5 equiv) was used without 2,2′-bipyridine

**Fig. 2** Mechanistic investigation. **a** The use of other difluorocarbene reagents for trifluoromethoxylation. **b** The identification of the oxygen source by [18]O-labeling. **c** The identification of the oxygen source by isolating $Ph_2S$. **d** The confirmation of the $AgOCF_3$ complex. [a]The optimal conditions are shown as Table 1, entry 21: substrate **1** (0.2 mmol), $Ph_3P^+CF_2CO_2^-$ (2.5 equiv), $Ph_2S=O$ (2.5 equiv), AgF (2 equiv), 2,2′-bipyridine (1.5 equiv), and 2,3,11,12-dibenzo-18-crown-6 (0.5 equiv) in THF (1.5 mL) at 60 °C for 0.5 h; [b]Yields were determined by [19]F NMR spectroscopy. [c]The [18]O content was determined by EI-MS. [d]Isolated yield calculated based on substrate **1–1**. [e]Isolated yield based on $Ph_2S=O$ consumed.

**Mechanistic investigations**. Further experimental evidence was collected to support the difluorocarbene oxidation process. The use of other difluorocarbene reagents such as $FSO_2CF_2CO_2TMS$[23] and $TMSCF_2Br$[8] could also give the desired trifluoromethoxylation product, albeit in a low yield, suggesting that difluorocarbene is a key intermediate (Fig. 2a). $CF_2=O$ could not be detected in the reaction mixtures, because it is a highly electrophilic species and would be rapidly attacked by AgF to provide $AgOCF_3$. Even stirring the mixture of $Ph_3P^+CF_2CO_2^-$ and $Ph_2S=O$ alone could not lead to the observation of $CF_2=O$, because $CF_2=O$ would easily react with the nucleophile, $Ph_3P$ generated from $Ph_3P^+CF_2CO_2^-$[9]. $Ph_2S=O$ should be the oxygen source to oxidize difluorocarbene to generate $CF_2=O$, since [18]O-labeled diphenyl sulfoxide afforded the $CF_3{}^{18}O$ product (Fig. 2b), and diphenyl sulfoxide underwent deoxygenation to afford diphenyl sulfide ($Ph_2S$) in a high yield based on $Ph_2S=O$ consumed (39% of $Ph_2S=O$ was recovered) (Fig. 2c) (Supplementary Methods). A stepwise reaction was performed to confirm the generation of the $AgOCF_3$ complex (Fig. 2d). Without the presence of a substrate, heating a mixture of $Ph_3P^+CF_2CO_2^-/Ph_2S=O/AgF$ with ligands at 60 °C for 0.5 h led to the formation of a number of unkonwn species, as detected by [19]F NMR spectroscopy (Supplementary Fig. 2). Two broad signals, appearing at −21.66 and −21.94 ppm in the [19]F NMR spectrum, respectively, may correspond to two different ligand-coordinated $AgOCF_3$ complexes[22]. Subsequent addition of substrate **1–1** afforded the desired trifluoromethoxylation product, further supporting that $AgOCF_3$ was generated from the $Ph_3P^+CF_2CO_2^-/Ph_2S=O/AgF$ system (Fig. 2d). In addition to the trifluoromethoxylation product, a fluorination byproduct was observed (Fig. 2d). However, almost no fluorination byproduct was observed under the optimal conditions (Table 1, entry 21), which suggests that $AgOCF_3$ was too reactive and decomposed easily.

DFT calculations at the M062X//6–31 + + G(d,p)/LANL2DZ level provided insights into the mechanism of the oxidation of difluorocarbene and the subsequent trifluoromethoxylation. We have previously demonstrated that $Ph_3P^+CF_2CO_2^-$ is an efficient difluorocarbene precursor, and has proposed that difluorocarbene is generated via a decarboxylation process, i.e., $Ph_3P^+CF_2CO_2^- \rightarrow Ph_3P^+CF_2^- \rightarrow :CF_2$[14,15,24]. Calculations revealed that the activation energy for this process is quite low (10.12 kcal mol$^{-1}$) (Supplementary Fig. 3 and Supplementary Data 1), supporting the mechanistic proposal. As an electron-deficient species, difluorocarbene can be attacked by $Ph_2S=O$ to form an $O-CF_2$ bond (Fig. 3, **INT-1**). The formation of this bond weakens the S–O bond in $Ph_2S=O$, as shown by the increasing S–O bond length from **TS-1** to **INT-1**. Back donation of the carbon lone pair strengthens the $O-CF_2$ bond and further weakens the S–O bond (Fig. 3, **TS-2**). Complete cleavage of the S–O bond releases $Ph_2S$ and carbonyl fluoride ($CF_2=O$), a process which is thermodynamically favored. $CF_2=O$ is electrophilic and is therefore trapped by AgF to generate $AgOCF_3$, which can readily convert the substrates to the final products. The Ag ion can activate the substrates by precipitating the AgBr salt. Identification of transition state **TS-2** enabled us to calculate the overall activation energy, i.e., 17.60 kcal mol$^{-1}$; this value is low and in agreement with the rapid process.

**The introduction of $CF_3O$ installation**. The above results revealed that difluorocarbene could indeed be oxidized to give carbonyl fluoride. The oxidation of difluorocarbene and the subsequent trifluoromethoxylation provides an efficient protocol for $CF_3O$ incorporation. $CF_3O$ incorporation has received increasing attention because the $CF_3O$ group is a common structural motif in pharmaceuticals[25,26], agrochemicals[27,28], and functional materials[29,30]. A number of effective trifluoromethoxylation methods have been developed, including nucleophilic[31–37],

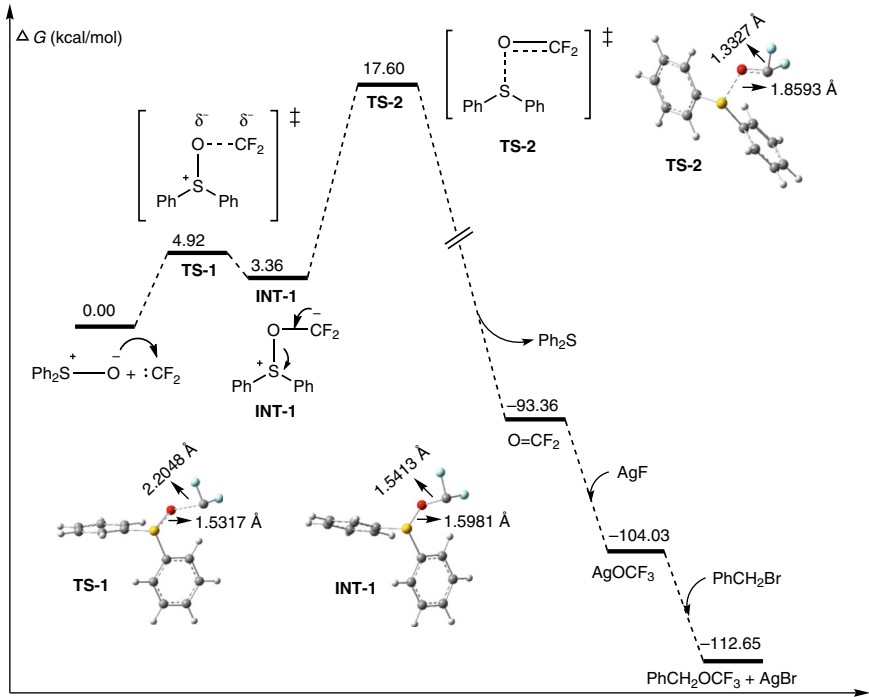

**Fig. 3** Relative free energies for difluorocarbene-oxidation-based trifluoromethoxylation. All calculations were performed in Gaussian 09 D01 package.

radical[38–40], and transition-metal-promoted[41–44] reactions. As the use of a $CF_3O$-containing reagent is required, these approaches cannot be directly applied to $^{18}O$-labeling trifluoromethoxylation. Furthermore, the $CF_3O$-containing reagents used are usually volatile, expensive, or difficult to prepare. In contrast, in the above protocol, $CF_3O$ moiety was formed from a reagent system consisting of $Ph_3P^+CF_2CO_2^-$, which could be easily prepared and easy-to-handle, an oxygen source and fluoride anion. Apparently, this reaction provides a strategy for $^{18}O$-labeling trifluoromethoxylation, which may be achieved by replacing the oxygen source with $^{18}O$-source. $^{18}O$-trifluoromethoxylation may show great value as $^{18}O$-labeling has found widespread application in various research areas such as proteomics[45–47] and synthetic chemistry[48–50].

**The substrate scope of trifluoromethoxylation.** Since difluorocarbene could be oxidized and the subsequent trifluoromethoxylation proceeded smoothly (Table 1, entry 21), we then investigated the substrate scope of trifluoromethoxylation. Figure 4 shows that electron-deficient, -neutral, and -rich benzyl bromides were all converted to the desired products in moderate to good yields (5–1 ~ 5–17). Various functional groups were tolerated, e.g., halide, ketone, ester, alkene, cyano, nitro, ether, and various heterocycles. Heterocycles usually have interesting physicochemical properties, and therefore the easy access to $CF_3O$-containing heterocycles could be useful in the life sciences (5–15 ~ 5–17). Transformation of secondary benzyl bromides gave moderate yields (5–18 ~ 5–22). The diphenyl substituted product (5–22) was unstable, and a heterolytic cleavage of the C–$OCF_3$ bond readily occurred to form a diphenyl-stabilized methyl cation, hydrolysis of which led to an alcohol by product ($Ph_2CH$-OH) in 35% isolated yield. In addition to benzyl bromides, allyl bromides were also converted under these conditions (5–23 ~ 5–28). The reactivity of alkyl bromide (5–29) was much lower than that of benzyl bromides. Alkyl iodides (5–30 ~ 5–33) underwent the desired reaction smoothly to give the expected products in moderate yields. A method for achieving direct access to a flavone derivative was developed (5–34) and a moderate yield

was obtained for a large-scale reaction (5–4), demonstrating the synthetic utility of this trifluoromethoxylation protocol.

Trioxsalen, a furanocoumarin and a psoralen derivative obtained from plants, can be used for phototherapy treatment of vitiligo and hand eczema[51]. A convenient route to the $CF_3O$-containing Trioxsalen derivative (8) was developed to further show the synthetic utility of this trifluoromethoxylation strategy. The trifluoromethoxylation of the precursor (7), prepared from the commercially available *m*-benzenediol by a reported procedure (Supplementary Fig. 1)[52,53], occurred smoothly to give the Trioxsalen derivative in a moderate yield (Fig. 5).

**$^{18}O$-Trifluoromethoxylation.** $^{18}O$-Labeling trifluoromethoxylation is challenging, because all reported trifluoromethoxylation methods have to use a $CF_3O$-containing reagent and the corresponding $CF_3^{18}O$-reagents are difficult to prepare. Recently, Tang used an $^{18}O$-labeled reagent, $ArSO_2–^{18}OCF_3$, to explore and elucidate the mechanism of the trifluoromethoxylation reaction; only a 33% $^{18}O$ content was obtained in the desired product[37]. They proposed that the low $^{18}O$-content was because of the $^{16}O$-$^{18}O$ exchange in the $SO_2–^{18}OCF_3$ moiety from the reagent. We employed $^{18}O$-labeled diphenyl sulfoxide ($Ph_2S=^{18}O$, $^{18}O$ content: 89%) as the oxygen source in this difluorocarbene-oxidation-based trifluoromethoxylation reaction. Since the reagent, $Ph_2S=^{18}O$, did not contain any $^{16}O$ atom, no $^{16}O$-$^{18}O$ exchange would occur and therefore the expected products were obtained with high $^{18}O$ contents (Fig. 6).

## Discussion
In summary, we have shown that difluorocarbene could be oxidized to afford carbonyl fluoride. This process was confirmed by the successful trifluoromethoxylation, $^{18}O$-trifluoromethoxylation, the observation of $AgOCF_3$ species, and DFT calculations. It is worth noting that the $^{18}O$-products were obtained with high $^{18}O$-contents. A $CF_3O$-containing Trioxsalen derivative was synthesized by this trifluoromethoxylation protocol. The oxidation of difluorocarbene may provide more possibilities for difluorocarbene chemistry.

**Fig. 4** Difluorocarbene-oxidation-based trifluoromethoxylation. Isolated yields are shown. Reaction conditions: substrate **1** (0.8 mmol), $Ph_3P^+CF_2CO_2^-$ (2.5 equiv), $Ph_2S=O$ (2.5 equiv), AgF (2 equiv), 2,2′-bipyridine (1.5 equiv), and 2,3,11,12-dibenzo-18-crown-6 (0.5 equiv) in THF (6 mL) at 60 °C for 0.5 h. [a]The yields in parentheses were determined by $^{19}F$ NMR spectroscopy. [b]0.2 mmol of substrate was used. [c]8 mmol of substrate was used.

## Methods

**Typical procedure for trifluoromethoxylation.** Into a 20 mL sealed tube were added benzyl bromide **1–1** (0.8 mmol, 197.7 mg, 1.0 equiv), $Ph_3P^+CF_2CO_2^-$ (2.0 mmol, 712.0 mg, 2.5 equiv), $Ph_2S=O$ (2.0 mmol, 404.6 mg, 2.5 equiv), AgF (1.6 mmol, 203.2 mg, 2.0 equiv), 2,2′-bipyridine (1.2 mmol, 187.4 mg, 1.5 equiv), 2,3,11,12-dibenzo-18-crown-6 (0.4 mmol, 144.2 mg, 0.5 equiv), and THF (6 mL) under a $N_2$ atmosphere. The tube was sealed and the reaction mixture was stirred at 60 °C for 30 min. After the mixture was cooled to room temperature, the pure product was isolated by flash column chromatography.

**Typical procedure for $^{18}O$-trifluoromethoxylation.** Into a 10-mL sealed tube were added benzyl bromide **1–1** (0.2 mmol, 49.4 mg, 1.0 equiv.), $Ph_3P^+CF_2CO_2^-$

(0.5 mmol, 178.0 mg, 2.5 equiv), $Ph_2S=^{18}O$ (0.5 mmol, 102.1 mg, 2.5 equiv), AgF (0.4 mmol, 51.0 mg, 2.0 equiv), 2,2′-bipyridine (0.3 mmol, 47.0 mg, 1.5 equiv), 2,3,11,12-dibenzo-18-crown-6 (0.1 mmol, 36.0 mg, 0.5 equiv), and THF (1.5 mL) under a $N_2$ atmosphere. The tube was sealed and the reaction mixture was stirred at 60 °C for 30 min, and the mixture was cooled to room temperature. The pure product was isolated by flash column chromatography, and the $^{18}O$ contents were determined by GC-MS (EI) spectroscopy.

For the preparation of starting materials and the characterization data of the products, see Supplementary Methods. For the NMR spectra of the compounds, see Supplementary Figs. 5–184. For EI spectra of the $^{18}O$-products, see Supplementary Figs. 185–214. For DFT calculations, see Supplementary Figs. 3 and 4 and Supplementary Data 1 and 2.

**Fig. 5** The synthesis of $CF_3O$-containing Trioxsalen derivative. The derivative was synthesized by a late-stage trifluoromethoxylation reaction.

**Fig. 6** Difluorocarbene-oxidation-based $^{18}O$-trifluoromethoxylation. Isolated yields. Reaction conditions: substrate **1** (0.2 mmol), $Ph_3P^+CF_2CO_2^-$ (2.5 equiv), $Ph_2S={}^{18}O$ (2.5 equiv), AgF (2 equiv), 2,2'-bipyridine (1.5 equiv), and 2,3,11,12-dibenzo-18-crown-6 (0.5 equiv) in THF (1.5 mL) at 60 °C for 0.5 h. The $^{18}O$ contents were determined by EI-MS.

## Data availability

The authors declare that the data supporting the findings of this study are available within the article and its Supplementary Information files or from the corresponding author on reasonable request.

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

## Acknowledgements

We thank the National Natural Science Foundation (21421002, 21672242, 21971252), Key Research Program of Frontier Sciences (CAS) (QYZDJSSW-SLH049), the Fujian Institute of Innovation, Chinese Academy of Sciences (FJCXY18040102) for financial

support. Computing resources were provided by the National Supercomputing Center of China in Shenzhen.

## Author contributions

J.Y. performed the experiments. D.Y. performed the DFT calculations. R.D. analyzed the data. J.-H.L. analyzed the data and wrote the manuscript. J.-C.X. designed the experiments and wrote the manuscript.

## Competing interests

The authors declare no competing interests.
