## [Peer Review File · Nature Communications]

Reviewers' comments:

Reviewer #1 (Remarks to the Author):

Trifluoromethoxy-molecules possess very specific properties which are very valuable in various fields of applications, from medicinal chemistry to materials.

Nevertheless, the synthesis of these compounds remains challenging, due to a crucial lack of reagents to perform trifluoromethoxylation.

Recently, some efficient reagents have been described but, because of their restricted scope of applications, the development of new methods is always desired.

In this paper, the authors develop an original oxidation reaction of easily accessible difluorocarbene to generate in situ the CF₃O⁻ anion which can then perform nucleophilic substitution.

Conceptually, this is a very innovative approach. Furthermore, with this method labeling with ¹⁸O could be performed, which constitutes the first efficient method to obtain [¹⁸O]CF₃O-molecules.

In their bibliography concerning nucleophilic trifluoromethoxylation (ref 28-31), some papers using CF₃OSO₂CF₃ and dinitrotrifluoromethoxybenzene to perform trifluoromethoxylation through nucleophilic substitutions, are lacking.

Reviewer #2 (Remarks to the Author):

Incorporation of a fluorine atom into organic molecules has attracted intensive attention because such a process often significantly alters the physicochemical properties of the parent molecules. In this report, Xiao et al. developed a novel strategy to install OCF₃ moiety via an S_N2 reaction mechanism using difluorocarbene precursors, diphenylsulfoxide, and alkyl bromides as starting materials. The novelty of this work includes oxidation of an electrophilic difluorocarbene species with sulfoxide, which allows access to ¹⁸O-labeled OCF₃ anion when ¹⁸O labeled diphenyl sulfoxide is used. The reaction takes advantage of the ambiphilic reactivity of difluorocarbene intermediate that turns from electrophilic to nucleophilic after reacting with a nucleophile. The resulting nucleophilic Ph₂S-OCF₂ intermediate would then eliminate the Ph₂S to give CF₂O. A similar protocol has been developed and reported by the authors using S₈, Se, and NH₃ as nucleophiles (ref. 13-17). Although Tang et al. (ref 31) were able to synthesize ¹⁸O-labeled OCF₃ containing molecules, the ¹⁸O-content was only 33% and with limited example. Using the method developed by Xiao, a wide range of alkyl ¹⁸OCF₃ compounds was obtained in moderate to good yields and with excellent content of ¹⁸O. Thus, this method could potentially be used to label organic molecules with ¹⁸O for biological studies, which is the significance of this report. The authors demonstrated that various alkyl, allylic, and benzylic halides could be used as viable substrates, and gram-scale synthesis of the desired products was feasible. Detailed experimental and computational studies were performed, and a reasonable process was proposed to explain the formation of the ¹⁸O-labeled OCF₃ anion.

Overall, this work would be of interest to both chemistry and biology communities. The strategy for accessing ¹⁸OCF₃ moiety is original, and the evidence presented in the manuscript is consistent with the conclusion. Thus, this manuscript will be suitable to be published in Nat. Commun. after the authors comment/address the following questions.

1. Have the authors tried a nucleophilic aromatic substitution reaction? If so, does it work?
2. Does the reaction tolerate complex bio-relevant molecules? If the authors could show the possibility of late-stage trifluoromethoxylation of complex molecules, it will further increase the significance of this work.
3. Crown ethers and bipyridine additives are critical for the reaction, what are their roles?

Reviewer #3 (Remarks to the Author):

I recommend reconsideration of this manuscript after adjustments to address concerns presented below.

The manuscript describes a trifluoromethoxylation reaction of benzylic, allylic, alkyl halides ($R-Br \diamond R-OCF_3$) by exploiting Xia and coworkers' $Ph_3P+CF_2CO_2^-$ reagent, along with Ph_2S+O^- and AgF . Reaction development and optimization is standard. The reaction proceeds in modest to moderate yields and has a reasonable scope, and the reaction can be adapted to generate ^{18}O -labeled derivatives. Computational studies suggest a mechanism that generates difluorocarbene ($:CF_2$), which subsequently undergoes attack by the sulfoxide, and subsequently decomposes to generate fluorophosgene ($F_2C=O$). Subsequently, the fluorophosgene reactions with AgF to generate $AgOCF_3$, which is the active species that reactions with the electrophile.

Overall, I find this quite an intriguing reaction that can convert common $R-Br$ reagents into a useful functional group, useful for applied chemists. Though yields could be improved, the reaction connects two useful functional groups and delivers products that have historically been quite challenging to accomplish. In this area, the deoxytrifluoromethoxylation reaction of Tang and co-workers probably leads the field (*ACIE*, 2018, 57, 292), and for practical synthetic purposes the present reaction is in the same classification. Differences in solvent and additives could be argued either way, so I don't think one reaction is intrinsically majorly more useful than the other. Most notably for synthetic purposes, the present reaction can generate ^{18}O labeled products with 90% incorporation of the label, which the Tang chemistry cannot accomplish. However, none of the labeled substrates in Scheme 4 are isolated, which I feel should be essential for *Nature Comm.* ^{19}F NMR yields can vary quite a bit relative to isolated yields, so I feel that these yields are inappropriate for a journal of this caliber.

Mechanistically, this sequence is reasonable, and a main innovative feature involves the oxidation/trapping of the difluorocarbene, and subsequent use to generate useful products, which I appreciate.

Other comments

Page 1, lines 24–27: these statements deserve references.

Page 2, lines 48–49: this statement is awkwardly placed, and I disagree that the overall optimization is evidence for oxidation of $:CF_2$. The mechanistic studies are better evidence for this process

Page 3, line 64: "Source" is misspelled.

Scheme 2: The reactions are quite cluttered, and it would be easier for readers/reviewers if there was sufficient vertical space between equations 1–4. Also, it is not clear what "optimal conditions" are for entries 1+3, relative to entry 2. Perhaps "optimal conditions" should be explicitly defined in the table footnotes. Also, eq 4 shows a major side product, but it is not discussed in the text.

Page 3, line 85: remove "highly".

Page 3, Figure 1 and computational discussion: typically, most synthetic chemistry journals use kcal/mol, not kJ/mol, and most of the synthetic community is calibrated for the former units. I strongly recommend converting all values to kcal/mol.

Page 4, line 100: Due to the challenges associated with ^{18}F labeling, speculation as to the ability of this reaction to accomplish such labelling is premature. I strongly recommend removing any suggestion that the present reaction could adequately deliver such probes without experimental evidence. Instead, the discussion of ^{18}O labeling is more appropriate.

Scheme 3: With modest yields, what are the major side products, or are there issues with conversion? This information would be helpful for any future users of the chemistry.

Page 4, line 114: Is there evidence for the methyl cation (5-22)? Could the desired product be detected transiently?

Page 5, lines 126–130: I recommend providing a more comprehensive comparison of the work by Tang vs. the present work. Specifically, the authors might explain why Tang's system only give 33% incorporation of ^{18}O (consider the preparation and composition of the reagent). Then a more clear rationale might be provided for why the present reaction has improved incorporation. Further, it might be interesting to suggest any other reaction that would potentially deliver such ^{18}O labeled substrates.

Synthetic procedures and SI characterization is acceptable (except for isolation of the ^{18}O labeled compounds, as mentioned above)

Response to Reviewers

Dear Reviewers:

Thank you for your valuable suggestions. We have addressed all comments by revising the manuscript and the Supporting Information files and also by providing a point-by-point response listed as follows. The changes in the Manuscript have been highlighted by a yellow background or by using the track changes mode in MS Word.

To Reviewer 1:

We thank Reviewer 1's support in the publication of the manuscript in *Nat Commun*.

Advice 1: In their bibliography concerning nucleophilic trifluoromethoxylation (ref 28-31), some papers using $\text{CF}_3\text{OSO}_2\text{CF}_3$ and dinitrotrifluoromethoxybenzene to perform trifluoromethoxylation through nucleophilic substitutions, are lacking.

Response: The papers on nucleophilic trifluoromethoxylation with $\text{CF}_3\text{OSO}_2\text{CF}_3$ (*Tetrahedron Lett* **49**, 449-454 (2008); *J Fluorine Chem* **131**, 200-207 (2010)) or dinitrotrifluoromethoxybenzene (*Adv Synth Catal* **352**, 2831-2837 (2010)) have been cited as Refs. 31, 32 and 33, respectively.

To Reviewer 2:

We thank Reviewer 1's support in the publication of the manuscript in *Nat Commun*.

Advice 1: Have the authors tried a nucleophilic aromatic substitution reaction? If so, does it work?

Response: Yew, we have tried the nucleophilic aromatic substitution. A variety of reaction conditions have been screened, including transition-metal-free, transition-metal-catalyzed, or stoichiometric-transition-metal-promoted conditions. For now, we have not established the optimal conditions yet.

Advice 2: Does the reaction tolerate complex bio-relevant molecules? If the authors could show the possibility of late-stage trifluoromethoxylation of complex molecules, it will further increase the significance of this work.

Response: Trioxsalen, a furanocoumarin and a psoralen derivative obtained from plants, can be used for phototherapy treatment of vitiligo and hand eczema. A convenient route to the CF_3O -containing Trioxsalen derivative was developed to show the synthetic utility of this trifluoromethoxylation strategy. The results have been included in the revised manuscript (Scheme 4).

Advice 3: Crown ethers and bipyridine additives are critical for the reaction, what are their roles?

Response: Crown ethers and bipyridine additives acted as ligands in the reactions. The reactions were performed at 60 °C, a warming condition that would readily lead to the decomposition of the key intermediate, AgOCF_3 , due to the high instability of this intermediate. Crown ethers and bipyridine additives could coordinate with the AgOCF_3 intermediate, increasing its stability and solubility. Therefore, the yields varied dramatically with or without these additives. The roles of these two additives are described in the DFT calculation section in Supporting Information file.

To Reviewer 3:

Reviewer 3's comments and suggestions are appreciated.

Advice 1: None of the labeled substrates in Scheme 4 are isolated, which I feel should be essential for Nature Comm. ^{19}F NMR yields can vary quite a bit relative to isolated yields, so I feel that these yields are inappropriate for at journal of this caliber.

Response: The ^{18}O -labeled products have been isolated. The isolated yields, which are only slightly different with ^{19}F NMR yields, are added in the revised manuscript.

Advice 2: Page 1, lines 24–27: these statements deserve references.

Response: The oxidation of a substrate is a process that involving the removal of electron(s) from the substrate. Therefore, electron-rich substrates would be easier to oxidize than electron-deficient substrates. For examples, electron-rich alkenes would more readily undergo epoxidation, and indoles, electron-rich heterocycles, are also prone to oxidation. Selected references are provided.

Due to the high electronegativity of fluorine element, difluorocarbene is an electron-deficient species and shows electrophilicity. The reference is added in the revised manuscript.

Advice 3: Page 2, lines 48–49: this statement is awkwardly placed, and I disagree that the overall optimization is evidence for oxidation of $:\text{CF}_2$. The mechanistic studies are better evidence for this process

Response: The statement at this place has been deleted.

Advice 4: Page 3, line 64: “Source” is misspelled.

Response: The spelling has been corrected.

Advice 5: Scheme 2: The reactions are quite cluttered, and it would be easier for readers/reviewers if there was sufficient vertical space between equations 1–4. Also, it is not clear what “optimal conditions” are for entries 1+3, relative to entry 2. Perhaps “optimal conditions” should be explicitly defined in the table footnotes. Also, eq 4 shows a major side product, but it is not discussed the text.

Response: The vertical space between equations 1–4 has been enlarged. The “optimal conditions” has been explicitly defined in the table footnotes. The major side product has been described in the text.

Advice 6: Page 3, line 85: remove “highly” .

Response: “highly” has been removed.

Advice 7: Page 3, Figure 1 and computational discussion: typically, most synthetic chemistry journals use kcal/mol, not kJ/mol, and most of the synthetic community is calibrated for the former units. I strongly recommend converting all values to kcal/mol.

Response: All values have been converted to kcal/mol.

Advice 8: Page 4, line 100: Due to the challenges associated with ^{18}F labeling, speculation as to the ability of this reaction to accomplish such labelling is premature. I strongly recommend removing any suggestion that the present reaction could adequately deliver such probes without experimental evidence. Instead, the discussion of ^{18}O labeling is more appropriate.

Response: The description about ^{18}F -labeling has been deleted.

Advice 9: Scheme 3: With modest yields, what are the major side products, or are there issues with conversion? This information would be helpful for any future users of the chemistry.

Response: In order to isolate a major side product, the trifluoromethoxylation of 4-phenylbenzyl bromide (4-PhC₆H₄CH₂Br, **1-1**) was re-examined. The Thin-layer chromatography (TLC) analysis of the reaction mixture revealed that the substrate was completely consumed. ¹⁹F NMR analysis of the reaction mixture showed that a major side product was HCF₃, which was formed via the reaction of difluorocarbene with fluoride anion. We have tried to isolate other side products by flash column chromatography, but complex mixtures were obtained. No major side product was isolated.

Advice 10: Page 4, line 114: Is there evidence for the methyl cation (**5-22**)? Could the desired product be detected transiently?

Response: After the reaction was finished, ¹⁹F NMR analysis revealed that the expected product (Ph₂CH-OCF₃, **5-22**) was produced in 44% yield. However, only trace amount of this product was isolated by flash column chromatography. Instead, an alcohol byproduct, Ph₂CH-OH, was isolated in 35% yield. This byproduct should be formed by hydrolysis of the methyl cation. The results have been included in the revised manuscript.

Advice 11: Page 5, lines 126 – 130: I recommend providing a more comprehensive comparison of the work by Tang vs. the present work. Specifically, the authors might explain why Tang's system only give 33% incorporation of ¹⁸O (consider the preparation and composition of the reagent). Then a more clear rationale might be provided for why the present reaction has improved incorporation. Further, it might be interesting to suggest any other reaction that would potentially deliver such ¹⁸O labeled substrates.

Response: The text has been revised. The reasons for the low ¹⁸O-content by Tang's method and the high ¹⁸O-content by our method have been provided in the revised manuscript.

REVIEWERS' COMMENTS:

Reviewer #2 (Remarks to the Author):

The authors have addressed all of my concerns. This manuscript can be published in Nat. Commun.

Reviewer #3 (Remarks to the Author):

The revised manuscript has undergone many changes to address all three reviewer critiques, and overall the manuscript is improved relative to the previous draft. The changes are generally well incorporated into the text (not just in the response letter), and overall the manuscript is now acceptable.

There are still some minor typos (Commas, misspelling, extra spaces, etc.) that might get improved during the standard editorial process.